# Incidence and risk of post-COVID-19 thromboembolic disease and the impact of aspirin prescription; nationwide observational cohort at the US Department of Veteran Affairs

**Anna D. Ware**[1]*, **Zachary P. Veigulis**[1,2], **Peter J. Hoover**[1], **Terri L. Blumke**[1], **George N. Ioannou**[3,4], **Edward J. Boyko**[5], **Thomas F. Osborne**[1,6]

1 National Center for Collaborative Healthcare Innovation, VA Palo Alto Healthcare System, Palo Alto, California, United States of America, 2 Department of Business Analytics, University of Iowa Tippie College of Business, Iowa City, Iowa, United States of America, 3 Center of Innovation for Veteran-Centered and Value-Driven Care, Veterans Affairs Puget Sound Healthcare System, Seattle, Washington, United States of America, 4 Division of Gastroenterology, Department of Medicine, University of Washington, Seattle, Washington, United States of America, 5 Seattle Epidemiologic Research and Information Center, Veterans Affairs Puget Sound Healthcare System, Seattle, Washington, United States of America, 6 Department of Radiology, Stanford University School of Medicine, Stanford, California, United States of America

* Anna.Ware@va.gov

## Abstract

### Introduction

COVID-19 triggers prothrombotic and proinflammatory changes, with thrombotic disease prevalent in up to 30% SARS-CoV-2 infected patients. Early work suggests that aspirin could prevent COVID-19 related thromboembolic disorders in some studies but not others. This study leverages data from the largest integrated healthcare system in the United States to better understand this association. Our objective was to evaluate the incidence and risk of COVID-19 associated acute thromboembolic disorders and the potential impact of aspirin.

### Methods

This retrospective, observational study utilized national electronic health record data from the Veterans Health Administration. 334,374 Veterans who tested positive for COVID-19 from March 2, 2020, to June 13, 2022, were included, 81,830 of whom had preexisting aspirin prescription prior to their COVID-19 diagnosis. Patients with and without aspirin prescriptions were matched and the odds of post-COVID acute thromboembolic disorders were assessed.

### Results

10.1% of Veterans had a documented thromboembolic disorder within 12 months following their COVID-19 diagnosis. Those with specific comorbidities were at greatest risk. Preexisting aspirin prescription was associated with a significant decrease risk of post-COVID-19

**Data Availability Statement:** Due to US Department of Veterans Affairs (VA) regulations

and our ethics agreements, the analytic data sets used for this study are not permitted to leave the VA firewall without a Data Use Agreement. This limitation is consistent with other studies based on VA data. However, VA data are made freely available to researchers with an approved VA study protocol. For more information, please visit https://www.virec.research.va.gov or contact the VA Information Resource Center at VIReC@va.gov.

**Funding:** The author(s) received no specific funding for this work.

**Competing interests:** The authors have declared that no competing interests exist.

thromboembolic disorders, including pulmonary embolism (OR [95% CI]: 0.69 [0.65, 0.74]) and deep vein thrombosis (OR [95% CI]: 0.76 [0.69, 0.83], but an increased risk of acute arterial diseases, including ischemic stroke (OR [95% CI]: 1.54 [1.46, 1.60]) and acute ischemic heart disease (1.33 [1.26, 1.39]).

## Conclusions

Findings demonstrated that preexisting aspirin prescription prior to COVID-19 diagnosis was associated with significantly decreased risk of venous thromboembolism and pulmonary embolism but increased risk of acute arterial disease. The risk of arterial disease may be associated with increased COVID-19 prothrombotic effects superimposed on preexisting chronic cardiovascular disease for which aspirin was already prescribed. Prospective clinical trials may help to further assess the efficacy of aspirin use prior to COVID-19 diagnosis for the prevention of post-COVID-19 thromboembolic disorders.

## Introduction

As of June 13th, 2023, an estimated 879,548 Veterans have been infected with SARS-CoV2, with 24,777 known COVID-19 deaths, and many suffering the long-term effects of the infection [1, 2]. Early in the pandemic, major health impacts of the disease were thought to be the consequence of the associated severe viral pneumonia [3, 4]. Since then, evidence demonstrates that COVID-19 triggers prothrombotic and proinflammatory changes throughout the body which can have multi-systemic adverse outcomes during and after acute infection [5–11]. In fact, multiple clinical trials and observational studies have documented thrombotic disease in 25–30% SARS-CoV-2 infected patients, primarily in seriously ill patients [12–19]. Other researchers have suggested that aspirin use could prevent post-COVID acute thromboembolic disorders, although results thus far have been mixed [20–22]. Without further evidence, current clinical guidelines recommend that patients diagnosed with COVID-19 do not initiate low-dose antiplatelet therapy (i.e., aspirin) for the prevention of thromboembolism associated with SARS-CoV-2 infection [23].

The Veterans Health Administration (VHA) is the largest integrated healthcare system in the United States (U.S.), providing care for over 9 million enrolled Veterans [24]. Veterans represent a statistically older population with multiple comorbidities and are therefore at a greater risk for COVID-19 complications and poorer outcomes compared to the general public [25]. The aims of this study were to 1) assess the incidence of post-acute COVID-19 associated thromboembolic disorders within our population, 2) identify risk factors for post-acute COVID-19 thromboembolic disorders, and 3) quantify the potential impact that preexisting aspirin use had on the risk of post-acute COVID-19 thromboembolic disorders.

## Materials and methods

### Data source and patients

We performed a retrospective observational analysis from March 2nd, 2020, until June 13th, 2023, of Veterans aged 18 years or older who were diagnosed with COVID-19. For this analysis, data was accessed from the VHA's electronic health record (EHR) database, the Corporate Data Warehouse (CDW) [26], between May 11th, 2023, and June 13th, 2023. Queries were developed to extract and collate patient demographic and clinical information.

Patients with a documented COVID-19 polymerase chain reaction (PCR) lab test from March 2nd, 2020, until June 13th, 2022, were identified. A 12-month follow-up period following the patient's first positive COVID-19 PCR test for incidence of post-COVID-19 acute thromboembolic disorders was included, making the full assessment timeline from March 2nd, 2020, until June 13th, 2023. Patients with a negative or not-detected test result, those with a non-PCR or antigen lab, or patients with an incomplete Care Assessment of Needs (CAN) score within 6 months prior to their COVID-19 PCR lab test were excluded from the analysis (cohort inclusion/exclusion flowchart depicted in Fig 1). The CAN score is automatically calculated weekly on all living Veterans who are actively assigned to primary care panels [27, 28], and was utilized to inform current health status as well as identify patients actively utilizing VA care.

## Variables

The first documented positive COVID-19 PCR test during the patient's latest episode of SARS-CoV2 infection was captured as the index date. For example, if a patient tested positive for COVID-19 in March 2020, then again in August 2021, with positive tests every two weeks through October 2021 for the 2nd infection, we captured the positive test in August 2021 as our index date.

Similar to methodologies established elsewhere [4, 29], VHA's EHR data was utilized to assess aspirin use. In short, aspirin use was defined as patients with an active aspirin prescription at the time of their index date. If a patient had no refills at this time, the prescription was only considered active if it was filled within 30 days prior to the index date. Patients who had an active non-VHA aspirin prescription documented in their EHR were also included in the aspirin cohort. All other patients in the cohort with no documented active aspirin prescription were classified as controls.

Thromboembolic disorders were captured using ICD-10 codes documented within the patient's EHR. This included venous thromboembolic disorders (pulmonary embolism (PE), deep vein thrombosis (DVT), cerebral venous sinus thrombosis (CVST), and other venous thromboembolic disorders), as well as arterial thromboembolic disorders (ischemic stroke, ischemic heart disease, and other arterial thromboembolic disorders). ICD-10 codes and their respective definitions can be found in S1 Table.

Additional covariates included age, sex, self-identified race or ethnicity, the CAN 1-Year Mortality score (Model Version 2.5), the Charlson Comorbidity Index (CCI), body mass index (BMI), common comorbidities, inpatient status at index date, COVID-19 vaccination status, the number of COVID-19 vaccine doses at the index date, and all-cause mortality within the follow-up period. The CAN score incorporates multiple structured data elements, including socio-demographics, diagnoses, vital signs, medications, lab values, and healthcare utilization data in its calculation and was utilized to quantify a patient's health risk. The score ranges from 0 to 99, with higher scores representing a greater risk of mortality [4, 26, 30].

## Statistical analysis

To assess post-COVID-19 acute thromboembolic disorders, contingency tables and unadjusted odds ratios were analyzed for any associations between the aspirin and control groups using the complete cohort. To mitigate potential confounding effects resulting from the retrospective observational nature of our dataset, propensity score matching (PSM) was utilized to match and compare patients in the aspirin vs control groups using the "nearest-neighbor" method within the RStudio "MatchIt" library (Version 3.6.2) [31] on the unscaled covariates of age, gender, and CAN score. Vaccination status was not included as a covariate in the matching algorithm since our assessment demonstrated that it was not significantly associated

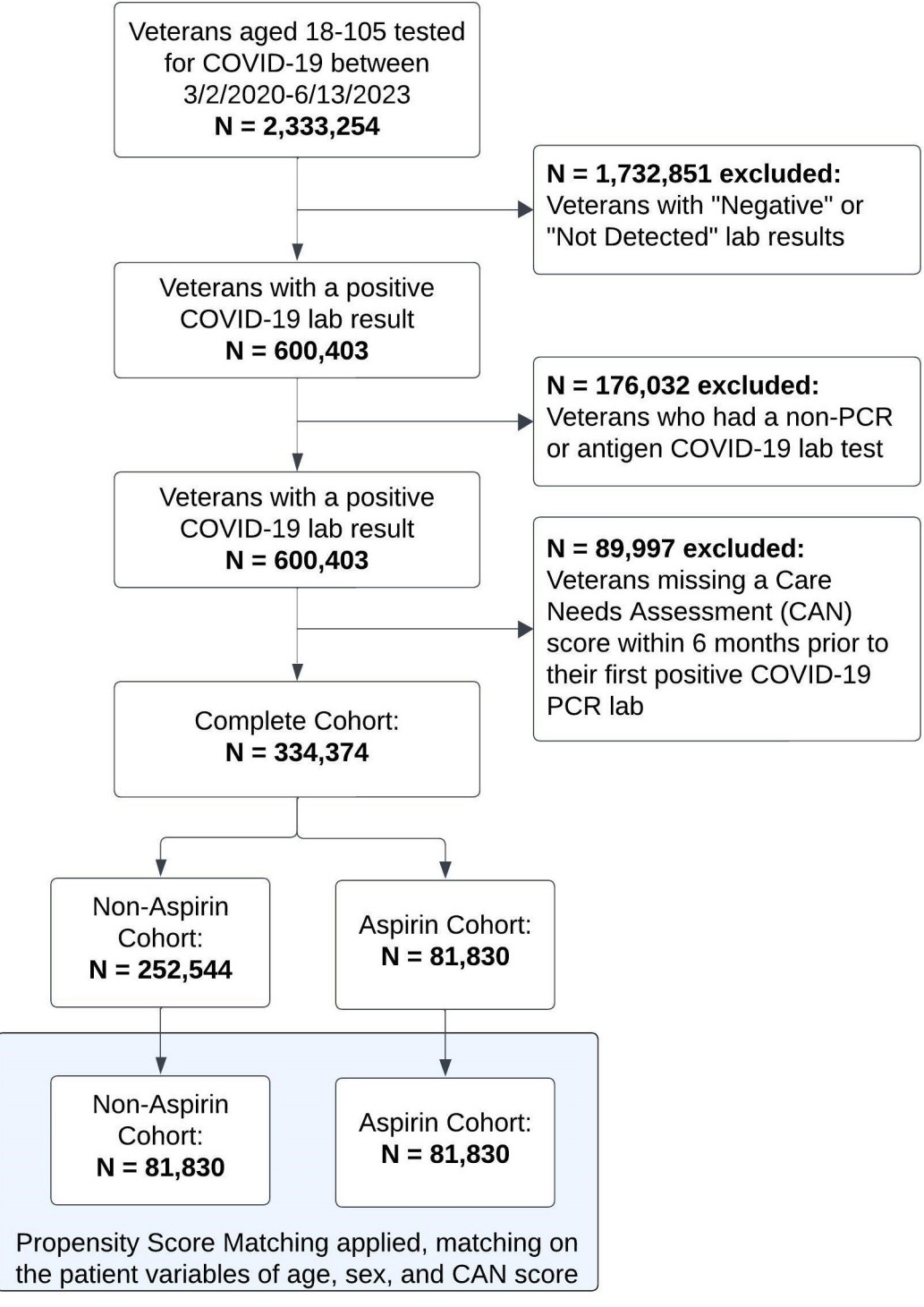

**Fig 1. Patient flowchart.**

with the treatment assignment nor the outcome. We then produced contingency tables and adjusted odds ratios for post-COVID-19 acute thromboembolic disorders. Odds ratios are reported as point estimates with 95% confidence intervals for the aspirin cohort with the control cohort as reference.

To identify Veterans at highest risk for post-COVID-19 acute thromboembolic disorders, we examined the association of each outcome subgroup using multivariate logistic regression models adjusting for the covariates of age, sex, BMI, CCI, race and/or ethnicity, and common comorbidities prior to the index date as a sub-analysis using the complete cohort. The CAN score was not included in this part of the analysis due to multicollinearity.

This study was approved by Stanford Institutional Review Board (Stanford University, Stanford, CA, USA; IRB #60725). Given the retrospective nature of our study and the minimal risk it posed to participants, the Stanford IRB granted this study a waiver for informed consent. This decision aligns with the ethical standards set forth in the 1964 Helsinki declaration and its later amendments, ensuring that all study procedures were conducted with the utmost respect for participant rights and privacy.

## Results

### Baseline patient characteristics

Of the 334,374 COVID-19 positive Veterans in our cohort, 24.5% had an active aspirin prescription prior to their index date (Fig 1). There was significant variation between groups at baseline in the complete cohort, demonstrating the need for PSM (Table 1). After PSM, 163,660 patients (81,830 aspirin users and non-users) were included in the analysis with an average (SD) age of 69.6 (10.8) years, 94.4% of the cohort were male, and average (SD) CAN score was 57.3 (26.7). Density plots of the matched covariates demonstrated optimal balance after PSM (S1 Fig).

### Risk factors for post-COVID-19 acute thromboembolic disorders

In our sub-analysis using the complete cohort prior to PSM, we found that those with any post-acute COVID-19 thromboembolic disorders had higher incidences of prior diagnoses compared to those without post-acute COVID-19 thromboembolic disorders. Specifically, these included congestive heart failure (CHF) (34.7% vs. 11.7%), chronic pulmonary disease (52.3% vs. 34.4%), chronic kidney disease (CKD) (35.4% vs. 14.2%), and hypertension (HTN) (89.7% vs. 58.8%; S2 Table). When assessing the odds of any thromboembolic disorders in the complete cohort, we observed an increased risk among older patients (OR [95% CI]: 1.03 [1.03, 1.03]), Black or African Americans (1.12 [1.08, 1.15]), males (1.26 [1.20, 1.33]), and for those diagnosed with CHF (1.45 [1.41, 1.50]), or HTN (1.56 [1.50, 1.62]). These increased risks were consistent across all subgroups of thromboembolic disorders (S2 Fig).

### Post-COVID-19 acute thromboembolic disorders

Prior to PSM, 10.1% of Veterans had documentation of any post-acute COVID-19 thromboembolic disease within the follow-up period (Table 2). This increased to 14.4% after matching on key baseline characteristics. The most prevalent among the case and control groups, respectively were ischemic stroke (6.9% vs. 4.6%, p<0.001), ischemic heart disease (4.8% vs. 3.7%, p<0.001), PE (2.1% vs. 3.0%, p<0.001), other venous thromboembolic disorders (2.1% vs. 2.8%, p<0.001), other arterial thromboembolic disorders (1.7% vs 1.3%, p<0.001), and DVT (1.1% vs. 1.3%, p<0.001).

### Aspirin's relationship with post-COVID-19 acute thromboembolic disorders

After PSM, aspirin use was associated with decreased risk of PE (OR [95% CI]: 0.69 [0.65, 0.74]), DVT (0.76 [0.69, 0.83]) and other venous thromboembolic disorders (0.73 [0.69, 0.78];

**Table 1. Patient characteristics before and after propensity score matching.**

| | Complete Cohort | | Propensity Score Matched Cohort* | |
|---|---|---|---|---|
| | Non-Users (N = 252544) | Aspirin Users (N = 81830) | Non-Users (N = 81830) | Aspirin Users (N = 81830) |
| **Matched Patient Characteristics** | | | | |
| Age (years), mean (SD) | 57.7 (16.4) | 69.6 (10.8) | 69.6 (10.8) | 69.6 (10.8) |
| Male | 217090 (86.0%) | 77245 (94.4%) | 77245 (94.4%) | 77245 (94.4%) |
| CAN 1 Year Mortality Score, mean (SD) | 38.8 (30.3) | 57.3 (26.7) | 57.3 (26.7) | 57.3 (26.7) |
| **Non-Matched Patient Characteristics** | | | | |
| **Race and/or Ethnicity** | | | | |
| Non-Hispanic White | 140166 (55.5%) | 50167 (61.3%) | 49468 (60.5%) | 50167 (61.3%) |
| Non-Hispanic Black or African American | 59295 (23.5%) | 18775 (22.9%) | 18778 (22.9%) | 18775 (22.9%) |
| Hispanic | 28784 (11.4%) | 6442 (7.9%) | 6673 (8.2%) | 6442 (7.9%) |
| Asian | 3767 (1.5%) | 660 (0.8%) | 643 (0.8%) | 660 (0.8%) |
| American Indian or Alaska Native | 2638 (1.0%) | 779 (1.0%) | 813 (1.0%) | 779 (1.0%) |
| Native Hawaiian/Other Pacific Islander | 2266 (0.9%) | 687 (0.8%) | 659 (0.8%) | 687 (0.8%) |
| Declined | 12178 (4.8%) | 3711 (4.5%) | 3864 (4.7%) | 3711 (4.5%) |
| Missing | 3450 (1.4%) | 609 (0.7%) | 932 (1.1%) | 609 (0.7%) |
| **Aspirin Dosage (mg/day), mean (SD)** | NA | 98.4 (61.9) | NA | 98.4 (61.9) |
| 81 mg/day | NA | 76976 (91.6%) | NA | 76976 (91.6%) |
| 162 mg/day | NA | 1172 (1.4%) | NA | 1172 (1.4%) |
| 243 mg/day | NA | 50 (0.1%) | NA | 50 (0.1%) |
| 325 mg/day | NA | 3022 (3.7%) | NA | 3022 (3.7%) |
| 405–650 mg/day | NA | 45 (0.0%) | NA | 45 (0.0%) |
| No Dose Listed (Non-VA prescription) | NA | 565 (0.7%) | NA | 565 (0.7%) |
| **CCI Score, mean (SD)** | 2.23 (2.90) | 4.14 (3.21) | 3.92 (3.22) | 4.14 (3.21) |
| **Body Mass Index (BMI), mean (SD)** | 30.8 (6.42) | 30.6 (6.24) | 29.9 (6.29) | 30.6 (6.24) |
| **Common Comorbidities Prior to COVID-19 Diagnosis** | | | | |
| Congestive Heart Failure | 26305 (10.4%) | 20604 (25.2%) | 14140 (17.3%) | 20604 (25.2%) |
| Chronic Pulmonary Disease | 82704 (32.7%) | 38334 (46.8%) | 35478 (43.4%) | 38334 (46.8%) |
| Chronic Kidney Disease | 33428 (13.2%) | 21198 (25.9%) | 17295 (21.1%) | 21198 (25.9%) |
| Diabetes | 32874 (13.0%) | 16500 (20.2%) | 13778 (16.8%) | 16500 (20.2%) |
| Hypertension | 134755 (53.4%) | 71040 (86.8%) | 59741 (73.0%) | 71040 (86.8%) |
| **Inpatient for COVID-19** | 78586 (31.1%) | 35212 (43.0%) | 35057 (42.8%) | 35212 (43.0%) |
| **Vaccinated status for COVID-19** | | | | |
| Unvaccinated at time of COVID diagnosis | 173230 (68.6%) | 55193 (67.4%) | 55693 (68.1%) | 56602 (67.4%) |
| Vaccinated at time of COVID diagnosis | 79314 (31.4%) | 26637 (32.6%) | 26137 (31.9%) | 27403 (32.6%) |
| **Number Vaccine Doses prior to COVID-19 Diagnosis** | | | | |
| Unvaccinated | 173230 (68.6%) | 55193 (67.4%) | 55693 (68.1%) | 55193 (67.4%) |
| 1 Dose | 8855 (3.5%) | 2157 (2.6%) | 2345 (2.9%) | 2157 (2.6%) |
| 2 Doses | 37887 (15.0%) | 10879 (13.3%) | 10869 (13.3%) | 10879 (13.3%) |
| 3 Doses | 32572 (12.9%) | 13601 (16.6%) | 12923 (15.8%) | 13601 (16.6%) |
| **Death within follow-up period** | 26254 (10.4%) | 7531 (9.2%) | 13890 (17.0%) | 7531 (9.2%) |

CAN = Care Assessment Needs, CCI = Charelson Comorbidity Index

*Propensity Score Matching was performed using the "nearest neighbor" method in R Studio (Version 3.6.2) through the "MatchIt" library, matching on the covariates of age, gender, and the VA's Care Assessment of Needs (CAN) 1-Year Mortality Score.

**Table 2. Frequency of post-COVID-19 acute thromboembolic disorders among aspirin users vs. non-users within 12 months following index date.**

| | Complete Unmatched Cohort | | | | PSM Matched Cohort* | | | |
|---|---|---|---|---|---|---|---|---|
| | Overall (N = 334,374) | Non- Aspirin (N = 252544) | Aspirin Users (N = 81830) | p-value[+] | Overall (N = 163660) | Non- Aspirin (N = 81830) | Aspirin Users (N = 81830) | p-value[+] |
| **Any Thromboembolic Disorders** | 33925 (10.1%) | 21233 (8.4%) | 12692 (15.5%) | <0.001 | 23643 (14.4%) | 10951 (13.4%) | 12692 (15.5%) | <0.001 |
| **Venous Thromboembolic Disorders** | | | | | | | | |
| Pulmonary Embolism | 6728 (2.0%) | 5021 (2.0%) | 1707 (2.1%) | 0.223 | 4148 (2.5%) | 2441 (3.0%) | 1707 (2.1%) | <0.001 |
| Deep Vein Thrombosis | 3206 (1.0%) | 2336 (0.9%) | 870 (1.1%) | 0.002 | 1949 (1.2%) | 1079 (1.3%) | 870 (1.1%) | <0.001 |
| Cerebral Venous Sinus Thrombosis | 75 (0.0%) | 53 (0.0%) | 22 (0.0%) | 0.619 | 39 (0.0%) | 17 (0.0%) | 22 (0.0%) | 0.726 |
| Other Venous Thromboembolic Disorders[1] | 6289 (1.9%) | 4594 (1.8%) | 1695 (2.1%) | <0.001 | 3947 (2.4%) | 2252 (2.8%) | 1695 (2.1%) | <0.001 |
| **Arterial Thromboembolic Disorders** | | | | | | | | |
| Ischemic Stroke | 12532 (3.7%) | 6867 (2.7%) | 5665 (6.9%) | <0.001 | 9432 (5.8%) | 3767 (4.6%) | 5665 (6.9%) | <0.001 |
| Ischemic Heart Disease (STEMI/NSTEMI) | 9626 (2.9%) | 5676 (2.2%) | 3950 (4.8%) | <0.001 | 6964 (4.3%) | 3014 (3.7%) | 3950 (4.8%) | <0.001 |
| Other Arterial Thromboembolic Disorders[2] | 3314 (1.0%) | 1911 (0.8%) | 1403 (1.7%) | <0.001 | 2472 (1.5%) | 1069 (1.3%) | 1403 (1.7%) | <0.001 |

*Propensity Score Matching (PSM) was performed using the "nearest neighbor" method in R Studio (Version 3.6.2) through the "MatchIt" library, matching on the covariates of age, gender, and the VA's Care Assessment of Needs (CAN) 1-Year Mortality Score.

[+]Chi-Sq and T-test used to determine significant associations between groups.

[1]Other Venous Thromboembolic Disorders includes acute embolism and thrombosis of the axillary, subclavian, internal jugular, superior vena cava, inferior vena cava, renal, and other thoracic veins. Also, acute embolism and thrombosis of other specified deep veins of lower extremities and veins of upper extremities.

[2]Other Arterial Thromboembolic Disorders includes both arterial embolism and thrombosis, and renal embolism and thrombosis.

Table 3 & Fig 2). However, there we observed an increased risk of post-COVID-19 arterial diseases, including ischemic stroke (OR [95% CI]: 1.54 [1.46, 1.60]), acute ischemic heart disease (1.33 [1.26, 1.39]), and other arterial thromboembolic disorders (1.32 [1.22, 1.43]).

## Discussion

COVID-19 can trigger systemic inflammatory and prothrombotic changes in the body, leading to negative adverse acute and long-term outcomes including cardiovascular disease [9–15]. As a result, researchers have speculated that aspirin could prevent some of the debilitating post-acute COVID-19 conditions such as thromboembolic disorders [5–8]. However, results have varied thus far and have been inconclusive [21, 22]. This study leveraged historical operational data from over 330,000 medical records, at the largest integrated healthcare system within the U.S., to help clarify associations.

We found that 10.1% of our cohort were diagnosed with any post-acute COVID-19 thromboembolic disorders within 12-months following their SARS-CoV2 infection. This rate is significantly greater than the expected baseline yearly incidence of thromboembolic disorders [32], yet similar to other reports of post COVID-19 thrombotic disease incidence [15]. Overall risk factors most strongly associated with post-COVID-19 acute thromboembolic disorders include older patients, males, those self-identified as Black or African American, and those with comorbidities such as CHF and HTN.

**Table 3. Unadjusted and adjusted odds ratios (95% confidence interval) for associations between patients who were prescribed aspirin at baseline and the occurrence of post-COVID-19 acute thromboembolic disorders within the follow-up period.**

| | Complete Unmatched Cohort (N = 334374) | PSM Matched Cohort* (N = 163660) |
|---|---|---|
| | Unadjusted ORs | Adjusted ORs |
| | (95% CIs) | (95% CIs) |
| **Any Thromboembolic Disorders** | **2.00 (1.95, 2.05)** | **1.19 (1.16, 1.22)** |
| **Venous Thromboembolic Disorders** | | |
| Pulmonary Embolism | 1.05 (0.99, 1.11) | **0.69 (0.65, 0.74)** |
| Deep Vein Thrombosis | **1.15 (1.06, 1.24)** | **0.76 (0.69, 0.83)** |
| Cerebral Venous Sinus Thrombosis | 1.28 (0.76, 2.08) | 1.05 (0.56, 1.99) |
| Other Venous Thromboembolic Disorders[1] | **1.14 (1.07, 1.20)** | **0.73 (0.69, 0.78)** |
| **Arterial Thromboembolic Disorders** | | |
| Ischemic Stroke | **2.66 (2.59, 2.78)** | **1.54 (1.47, 1.60)** |
| Acute Ischemic Heart Disease (STEMI/NSTEMI) | **2.21 (2.12, 2.30)** | **1.33 (1.26, 1.39)** |
| Other Arterial Thromboembolic Disorders[2] | **2.29 (2.13, 2.45)** | **1.32 (1.22, 1.43)** |

Significant odds ratios (OR) are bolded (p<0.05

*Propensity Score Matching (PSM) was performed using the "nearest neighbor" method in R Studio (Version 3.6.2) through the "MatchIt" library, matching on the covariates of age, gender, and the VA's Care Assessment of Needs (CAN) 1-Year Mortality Score.

**Adjusted for CAN 1-Year Mortality Score, Male, and Age in models prior to PSM matching

[1]Other Venous Thromboembolic Disorders includes acute embolism and thrombosis of the axillary, subclavian, internal jugular, superior vena cava, inferior vena cava, renal, and other thoracic veins. Also, acute embolism and thrombosis of other specified deep veins of lower extremities and veins of upper extremities.

[2]Other Arterial Thromboembolic Disorders includes arterial embolism and thrombosis and renal embolism and thrombosis.

Interestingly, our findings demonstrate contrasting associations with venous and arterial thromboembolic disease risk for those taking aspirin vs those not taking aspirin prior to being diagnosed with COVID-19. Specifically, COVID-19 positive Veterans with preexisting aspirin prescriptions had a decreased risk of venous thromboembolic disorders, but an increased risk

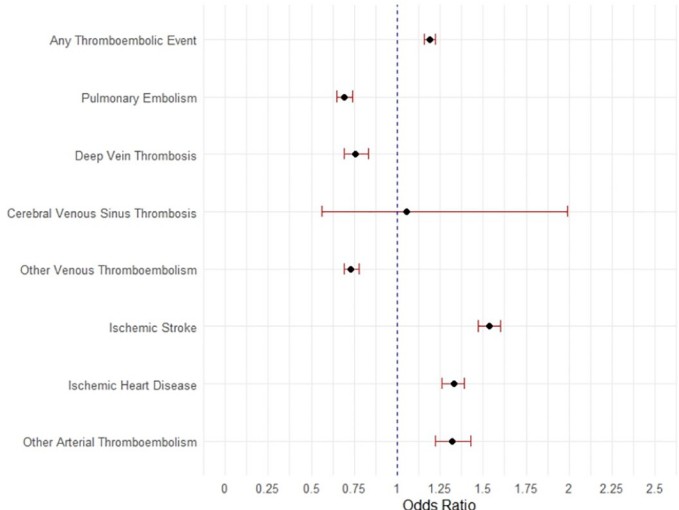

**Fig 2. Forest plot displaying the association (odds ratio [95% CI]) between patients who were prescribed aspirin at baseline and the occurrence of post-COVID-19 acute thromboembolic disorders within the follow-up period among the propensity score matched cohort (N = 168010).**

of arterial thromboembolic diseases during the follow-up period. Since daily aspirin use is typically prescribed as a prophylaxis for arterial thromboembolic disease [33], our observed arterial disease results could potentially reflect the biased baseline population risks in the aspirin group. Therefore, Veterans who were predetermined by their clinician to be at increased risk for arterial disease and were prophylactically taking aspirin to reduce that risk might have a superimposed higher risk during a prothrombotic state from COVID-19 that the antithrombotic effects of aspirin could not overcome.

Strengths of this study include that our analysis was performed on a large population-based assessment of Veterans, utilizing data from a nationwide integrated healthcare system, in which diagnoses, medication use, and laboratory studies are all observable. VHA is in a unique position to assess OTC medications because VHA can directly provide Veterans with OTC medications and systematically documents all active medications for those who acquire OTCs elsewhere. Additionally, while a handful of previous studies attempted to assess the effects of aspirin on post-COVID thromboembolic disorders [21, 22], there were limitations in the ability to distinguish between historical, chronic, and acute aspirin uses. In contrast, our assessment presents associations for our cohort who were prescribed daily aspirin prior to their COVID-19 diagnosis, and not as a new or historical-inactive treatment.

Our results are limited as our Veteran population may face unique Veteran related health challenges [1], therefore, these associations may not be generalizable to non-VHA settings. Although our PSM models demonstrated balance across covariates, they were not able to eliminate the expected confounding that was related to the association of daily aspirin being prescribed to those who are clinically determined to be at higher risk for cardiovascular disease. As this is a retrospective evaluation, we cannot establish direct cause and effect, and therefore future randomized-control trials required to assess the potential of aspirin as a drug repurposing option to prevent post-COVID acute thromboembolic disorders.

## Conclusion

Preexisting aspirin prescription at the time of COVID-19 diagnosis was associated with significantly decreased risk of venous thromboembolic disorders and an increased risk of arterial thromboembolic disease. The risk of arterial disease may, in part, be associated with preexisting arterial disease that triggered the historical daily aspirin prescription before COVID-19 diagnosis. Prospective clinical trials are required to assess the efficacy and appropriateness of aspirin use prior to COVID-19 diagnosis for the prevention of acute post-COVID-19 thromboembolic disorders.

## Supporting information

**S1 Fig. Density plots displaying the balance on the matched covariates of age, gender, and the VA's Care Assessment of Needs (CAN) 1-Year Mortality score before and after propensity score matching using the "nearest neighbor" method in R Studio (Version 3.6.2).** (DOCX)

**S2 Fig. Subgroup analyses of the odds of post-COVID-19 acute thromboembolic disorders using the complete cohort (N = 334,374).** (DOCX)

**S1 Table. ICD-10 Codes and definitions for acute thromboembolic disorders included in the analysis.** (DOCX)

**S2 Table. Sub-analysis cohort characteristics among those with and without thromboembolic disorders within 12 months following the patient's index date using the complete cohort (N = 334,374).**
(DOCX)

## Acknowledgments

**Disclaimer**: The findings and conclusions in this report are those of the authors and do not necessarily represent the official position of the US Department of Veterans Affairs.

## Author Contributions

**Conceptualization:** Anna D. Ware, Zachary P. Veigulis, Terri L. Blumke, George N. Ioannou, Edward J. Boyko, Thomas F. Osborne.

**Data curation:** Anna D. Ware, Zachary P. Veigulis, Peter J. Hoover, Terri L. Blumke.

**Formal analysis:** Anna D. Ware, Zachary P. Veigulis, Peter J. Hoover, Terri L. Blumke, George N. Ioannou, Edward J. Boyko, Thomas F. Osborne.

**Investigation:** Anna D. Ware, Zachary P. Veigulis.

**Methodology:** Anna D. Ware, Zachary P. Veigulis.

**Project administration:** Anna D. Ware.

**Resources:** Anna D. Ware.

**Software:** Anna D. Ware.

**Supervision:** Thomas F. Osborne.

**Validation:** Anna D. Ware.

**Visualization:** Anna D. Ware.

**Writing – original draft:** Anna D. Ware, Zachary P. Veigulis, Peter J. Hoover, Terri L. Blumke, George N. Ioannou, Edward J. Boyko, Thomas F. Osborne.

**Writing – review & editing:** Anna D. Ware, Peter J. Hoover, Terri L. Blumke, George N. Ioannou, Edward J. Boyko, Thomas F. Osborne.

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
