## [Decision Letter · Decision Letter 0]

27 May 2024

PONE-D-24-13134Incidence and Risk of Post-COVID-19 Thromboembolic Disease and the Impact of Aspirin Prescription; Nationwide Observational Cohort at the US Department of Veteran Affairs.PLOS ONE

Dear Dr. Ware,

Thank you for submitting your manuscript to PLOS ONE. After careful consideration, we feel that it has merit but does not fully meet PLOS ONE’s publication criteria as it currently stands. Therefore, we invite you to submit a revised version of the manuscript that addresses the points raised during the review process.

The manuscript is overall well written. Only minor revisions are required.

We look forward to receiving your revised manuscript.

Kind regards,

Raffaele Serra, M.D., Ph.D

Academic Editor

PLOS ONE

Additional Editor Comments:

The manuscript requires only minor revisions.

Reviewers' comments:

Reviewer's Responses to Questions

**Comments to the Author**

1. Is the manuscript technically sound, and do the data support the conclusions?

Reviewer #1: Yes

2. Has the statistical analysis been performed appropriately and rigorously? 

Reviewer #1: Yes

3. Have the authors made all data underlying the findings in their manuscript fully available?

Reviewer #1: Yes

4. Is the manuscript presented in an intelligible fashion and written in standard English?

Reviewer #1: No

5. Review Comments to the Author

Reviewer #1: This study leverages data from the largest integrated healthcare system in the United States to better understand this association. I suggest to include in the references the following paper : Ielapi N, et al. Malattia cardiovascolare come biomarker per un aumento del rischio di infezione da COVID-19 e relativa prognosi sfavorevole. BioMark Med. 2020 giugno;14(9):713-716. doi: 10.2217/bmm-2020-0201, because it is in line with the issue of this paper. I suggest a extensive revision of the language.

6. PLOS authors have the option to publish the peer review history of their article (what does this mean?). If published, this will include your full peer review and any attached files.

Reviewer #1: No

---

## [Author Response · Author response to Decision Letter 0]

29 May 2024

PLOS ONE 

May 29, 2024

RE: Manuscript Number: PONE-D-24-13134

Dear Editors,

Thank you for the opportunity to revise our manuscript entitled, “Incidence and Risk of Post-COVID-19 Thromboembolic Disease and the Impact of Aspirin Prescription; Nationwide Observational Cohort at the US Department of Veteran Affairs.” for publication in the PLOS ONE. We sincerely appreciate the reviewers’ comments and have made considerable efforts to address them. Please see the table below for a point-by-point response to reviewer. 

Reviewer 1 Comments: This study leverages data from the largest integrated healthcare system in the United States to better understand this association. I suggest to include in the reference the following paper: Ielapi N, et al. Malattia cardiovascolare come biomarker per un aumento del rischio di infezione da COVID-19 e relativa prognosi sfavorevole. BioMark Med. 2020 giugno;14(9):713-716. doi: 10.2217/bmm-2020-0201, because it is in line with the issue of this paper. I suggest a extensive revision of the language.

Authors Response: Thank you for this assessment of our work. We appreciate your thoughtful and thorough suggestions. I have added the following reference to the introduction of our paper following the sentence, “Since then, evidence demonstrates that COVID-19 triggers prothrombotic and proinflammatory changes throughout the body which can have multi-systemic adverse outcomes during and after acute infection” on page 4 as I concur that it is pertinent to this manuscript: 

• Ielapi N, Licastro N, Provenzano M, Andreucci M, Franciscis S, Serra R. Cardiovascular disease as a biomarker for an increased risk of COVID-19 infection and related poor prognosis. Biomark Med. 2020 Jun;14(9):713-716. doi: 10.2217/bmm-2020-0201. Epub 2020 May 19. PMID: 32426991; PMCID: PMC7236792.

Additionally, I have re-read through the manuscript and edited the language to be more clear, correct, and unambiguous. 

Specifically:

• In the introduction on page 4, line 82: 

o “Other researcher suggest that aspirin…” has been changed to: “Other researchers have suggested that aspirin…”

• In the introduction on page 4, lines 90-92:

o “The aims of this study were to 1) assess the incidence of COVID-19 associated acute thromboembolic disorders…” has been adjusted to state: “The aims of this study were to 1) assess the incidence of post-acute COVID-19 associated thromboembolic disorders within our population…”

• In the introduction on page 4, lines 94-96:

o “3) quantify the potential impact that preexisting aspirin use had on the risk of thromboembolic disease.” has been modified to state “3) quantify the potential impact that preexisting aspirin use had on the risk of post-acute COVID-19 thromboembolic disorders.”

• In the methods section on page 5, line 111:

o “…weekly on all living Veterans assigned to primary care panels.” was modified to: “…weekly on all living Veterans who are actively assigned to primary care panels.”

• In the methods section on page 6, line 120-122:

o “… the prescription was only considered active if it was filled up to 30 days prior to the index date.” was modified to state: “…the prescription was only considered active if it was filled within 30 days prior to the index date.”

• In the methods section on page 6, line 125:

o “Thromboembolic disorders were captured using ICD-10 codes within the patient’s EHR.” was edited to “Thromboembolic disorders were captured using ICD-10 codes documented within the patient’s EHR.”

• In the methods section on page 6, line 132:

o “…vaccination status and the number of vaccine doses at the index date, and all-cause mortality within the follow-up period.” was modified to add clarification: “…COVID-19 vaccination status, the number of COVID-19 vaccine doses at the index date, and all-cause mortality within the follow-up period.”

• In the results section on page 7, lines 164-166: 

o “Of the 334,374 COVID-19 Veterans in our cohort, 24.5% had an active aspirin prescription prior to their index date.” we added clarification on the cohort by stating “Of the 334,374 COVID-19 positive Veterans in our cohort, 24.5% had an active aspirin prescription prior to their index date.”

• In the results section on page 8, lines 167-168:

o “After PSM, 163,660 patients (81,830 aspirin users and non-users) were included in the analysis with an average (SD) age was 69.6 (10.8) years, 94.4% were male, and average (SD) CAN score was 57.3 (26.7)” was modified to state “After PSM, 163,660 patients (81,830 aspirin users and non-users) were included in the analysis with an average (SD) age of 69.6 (10.8) years, 94.4% of the cohort were male, and average (SD) CAN score was 57.3 (26.7).”

• In the results section on page 8, lines 173-181:

o “In our sub-analysis using the complete cohort prior to PSM, those with an incidence of any post-acute COVID-19 thromboembolic disorders vs not, were prior diagnoses of congestive heart failure (CHF) (34.7% vs. 11.7%), chronic pulmonary disease (52.3% vs. 34.4%), CKD (35.4% vs. 14.2%), and hypertension (HTN) (89.7% vs. 58.8%; Supp Table 2). When assessing the odds of any thromboembolic disorders in the complete cohort, there was an increased risk among older patients (OR [95% CI]: 1.03 [1.03, 1.03]), Black or African Americans (1.12 [1.08, 1.15]), male (1.26 [1.20, 1.33]), and for those diagnosed with CHF (1.45 [1.41, 1.50]), or HTN (1.56 [1.50, 1.62]) across all subgroups of thromboembolic disorders (Supp Figure 2).” was modified to state: “In our sub-analysis using the complete cohort prior to PSM, we found that those with any post-acute COVID-19 thromboembolic disorders had higher incidences of prior diagnoses compared to those without post-acute COVID-19 thromboembolic disorders. Specifically, these included congestive heart failure (CHF) (34.7% vs. 11.7%), chronic pulmonary disease (52.3% vs. 34.4%), chronic kidney disease (CKD) (35.4% vs. 14.2%), and hypertension (HTN) (89.7% vs. 58.8%; Supp Table 2). When assessing the odds of any thromboembolic disorders in the complete cohort, we observed an increased risk among older patients (OR [95% CI]: 1.03 [1.03, 1.03]), Black or African Americans (1.12 [1.08, 1.15]), males (1.26 [1.20, 1.33]), and for those diagnosed with CHF (1.45 [1.41, 1.50]), or HTN (1.56 [1.50, 1.62]). These increased risks were consistent across all subgroups of thromboembolic disorders (Supp Figure 2).”

• In the results section on page 8, lines 173-182:

o “Prior to PSM, 10.1% of Veterans had a condition typically related to thromboembolic disease within the follow-up period.” was modified to: “Prior to PSM, 10.1% of Veterans had documentation of any post-acute COVID-19 thromboembolic disease within the follow-up period (Table 2).”

• In the discussion section on page 9, lines 210-219:

o “Interestingly, our findings demonstrate opposite venous and arterial thromboembolic disease risk associations for those taking aspirin vs not prior to being diagnosed with COVID-19… Because daily aspirin use is typically prescribed as a prophylaxis for arterial thromboembolic disease, our observed arterial disease results could potentially reflect the biased baseline population risks in the aspirin group. Therefore, Veterans who were predetermined to be at increased risk for arterial disease by their clinician and prophylactically taking aspirin to reduce that risk may have a superimposed higher risk during a prothrombotic state from COVID-19 that the antithrombotic effects of aspirin could not overcome.” was modified to state: “Interestingly, our findings demonstrate contrasting associations with venous and arterial thromboembolic disease risk for those taking aspirin vs those not taking aspirin prior to being diagnosed with COVID-19….Since daily aspirin use is typically prescribed as a prophylaxis for arterial thromboembolic disease, our observed arterial disease results could potentially reflect the biased baseline population risks in the aspirin group. Therefore, Veterans who were predetermined by their clinician to be at increased risk for arterial disease and were prophylactically taking aspirin to reduce that risk might have a superimposed higher risk during a prothrombotic state from COVID-19 that the antithrombotic effects of aspirin could not overcome.”

• In the discussion section on page 9, line 234:

o “Strengths of this study include a large population-based assessment of Veterans…”was modified to state: “Strengths of this study include that our analysis was performed on a large population-based assessment of Veterans…”

I hope the manuscript will meet the high standards of the PLOS ONE. We look forward to hearing from you at your earliest convenience.

Best regards,

Anna Ware, MPH, MS

---

## [Editor Report · Decision Letter 1]

26 Jun 2024

Incidence and Risk of Post-COVID-19 Thromboembolic Disease and the Impact of Aspirin Prescription; Nationwide Observational Cohort at the US Department of Veteran Affairs.

PONE-D-24-13134R1

Dear Dr. Ware,

We’re pleased to inform you that your manuscript has been judged scientifically suitable for publication and will be formally accepted for publication once it meets all outstanding technical requirements.

Kind regards,

Raffaele Serra, M.D., Ph.D

Academic Editor

PLOS ONE

Additional Editor Comments (optional):

amended manuscript is acceptable.
---

## [Editor Report · Acceptance letter]

2 Jul 2024

PONE-D-24-13134R1 

PLOS ONE

Dear Dr. Ware, 

I'm pleased to inform you that your manuscript has been deemed suitable for publication in PLOS ONE. Congratulations! Your manuscript is now being handed over to our production team.

Kind regards, 

on behalf of

Prof. Raffaele Serra 

Academic Editor

PLOS ONE